# Alpha-Tocopherol Metabolites (The Vitamin E Metabolome) and Their Interindividual Variability during Supplementation

**DOI:** 10.3390/antiox10020173

**Published:** 2021-01-25

**Authors:** Desirée Bartolini, Rita Marinelli, Danilo Giusepponi, Roberta Galarini, Carolina Barola, Anna Maria Stabile, Bartolomeo Sebastiani, Fabiola Paoletti, Michele Betti, Mario Rende, Francesco Galli

**Affiliations:** 1Department of Pharmaceutical Sciences, University of Perugia, 06126 Perugia, Italy; ritamarinelli93@gmail.com (R.M.); francesco.galli@unipg.it (F.G.); 2Department of Medicine, University of Perugia, 06126 Perugia, Italy; anna.stabile@unipg.it (A.M.S.); mario.rende@unipg.it (M.R.); 3Istituto Zooprofilattico Sperimentale dell’Umbria e delle Marche “Togo Rosati”, 06126 Perugia, Italy; d.giusepponi@izsum.it (D.G.); r.galarini@izsum.it (R.G.); c.barola@izsum.it (C.B.); f.paoletti@izsum.it (F.P.); 4Department of Chemistry, Biology and Biotechnology, University of Perugia, 06126 Perugia, Italy; bartolomeo.sebastiani@unipg.it; 5Department of Biomolecular Sciences, University of Urbino “Carlo Bo”, 61029 Urbino, Italy; michele.betti@uniurb.it

**Keywords:** α-tocopherol, vitamin E, metabolomics, nutrigenomics, pregnane X receptor, lipoxygenase-5, peroxisome proliferator-activated receptor-γ, mass spectrometry, interindividual variability

## Abstract

The metabolism of α-tocopherol (α-TOH, vitamin E) shows marked interindividual variability, which may influence the response to nutritional and therapeutic interventions with this vitamin. Recently, new metabolomics protocols have fostered the possibility to explore such variability for the different metabolites of α-TOH so far identified in human blood, i.e., the “vitamin E metabolome”, some of which have been reported to promote important biological functions. Such advances prompt the definition of reference values and degree of interindividual variability for these metabolites at different levels of α-TOH intake. To this end, a one-week oral administration protocol with 800 U RRR-α-TOH/day was performed in 17 healthy volunteers, and α-TOH metabolites were measured in plasma before and at the end of the intervention utilizing a recently validated LC-MS/MS procedure; the expression of two target genes of α-TOH with possible a role in the metabolism and function of this vitamin, namely pregnane X receptor (PXR) and the isoform 4F2 of cytochrome P450 (CYP4F2) was assessed by immunoblot in peripheral blood leukocytes. The levels of enzymatic metabolites showed marked interindividual variability that characteristically increased upon supplementation. With the exception of α-CEHC (carboxy-ethyl-hydroxychroman) and the long-chain metabolites M1 and α-13′OH, such variability was found to interfere with the possibility to utilize them as sensitive indicators of α-TOH intake. On the contrary, the free radical-derived metabolite α-tocopheryl quinone significantly correlated with the post-supplementation levels of α-TOH. The supplementation stimulated PXR, but not CYP4F2, expression of leucocytes, and significant correlations were observed between the baseline levels of α-TOH and both the baseline and post-supplementation levels of PXR. These findings provide original analytical and molecular information regarding the human metabolism of α-TOH and its intrinsic variability, which is worth considering in future nutrigenomics and interventions studies.

## 1. Introduction 

The term vitamin E refers to the essential micronutrient α-tocopherol (α-TOH) (Figure 1) and, at the same time, to a group of plant-derived toco-chromanols (tocopherols and tocotrienols), which, despite having many biological properties, do have very poor, or rather absent, vitamin function [1].

α-TOH is a fat-soluble H atom donor and chain breaker of lipid peroxidation reactions [2]. As a consequence, the supplementation of α-TOH has extensively been investigated with the aim of combat the oxidative stress associated with age-related and inflammatory ailments, including cancer, cardiovascular, and neurodegenerative diseases [3,4,5]. 

However, nutritional and therapeutic applications of α-TOH are hardly affected by the biological heterogeneity that characterizes the metabolism and function of this vitamin in humans. Marked interindividual variability is characteristically observed in the vitamin levels of healthy subjects [6,7]. Heterogeneities have also been described in the biological response to α-TOH, including its effect of protection against lipid peroxidation [8,9], and may help to explain the inconsistent results obtained in most of the clinical trials carried out so far on this vitamin [10,11]. 

Along with variations in the dietary intake of the vitamin, genetic background and specific organ dysfunction (for example, subclinical defects of the gastrointestinal tract associated with malabsorption or defects of the liver function) may help to explain individual differences in the levels and metabolism of α-TOH. 

Effective strategies of nutritional assessment to address such variability are lacking. Former studies [6], pointed to the existance at the individual level of different abilities in absorbing the ingested vitamin and in the hepatic expression of α-TOH binding proteins, such as the α-TOH transport protein (α-TTP), which are rather difficult aspects to evaluate in humans. Furthermore, α-TTP and other α-TOH binding proteins associated with the cellular trafficking of the vitamin, such as the human α-tocopherol associated protein (hTAP) [12], present allelic variants associated with lower vitamin concentrations in plasma at baseline and post-supplementation [13]. The cytochrome P450 family (CYPs) of isoenzymes involved in the initial step of the enzymatic metabolism of vitamin E, i.e., the ω-hydroxylation of the side chain, may also play a role as a factor of biological variability. These isoenzyme include CYP3A4 and CYP4F2 [14,15], which are encoded by highly polymorphic genes [16]. Moreover, their expression depends on the activity of receptor-dependent transcription factors that are affected by a wide range of environmental substrates, and physiological and genetic factors [17,18], as well as by potential interactions with drug therapies and other fat-soluble vitamins, such as vitamin K [19].

These differences in the uptake and molecular regulation of the enzymatic metabolism of vitamin E, may also explain the marked variability of urine and plasma levels of carboxy-ethyl-hydroxychroman (CEHC) metabolites observed among healthy subjects (Figure 1) [20,21,22]; these are the final products in the enzymatic catabolism of this vitamin and are considered to represent good biomarkers of its intake [23]. Kelly et al. first investigated with a systematic approach the concentrations of α-TOH in human plasma and those of α-CEHC and quinone lactone (QL) metabolite in urine after α-TOH supplementation of healthy subjects [21]; what they found, besides the expected interindividual variability, was a high repeatability and correlation of these parameters in the study participants, the correspondence of which was confirmed over a period of 1 year. The stability of this phenotype suggests that the different aspects that control the uptake and biotransformation of α-TOH (reviewed elsewhere in Reference [24]) are under genetic regulation.

In recent times, long-chain metabolites (LCMs; Figure 1) of vitamin E have come to the attention of the scientific community. A series of studies provided the technology to synthesize and determine these metabolites in biological systems [23,25], and others identified their anti-inflammatory, anti-atherogenic, and detoxification properties that are superior to those of their vitamin precursors (reviewed in References [26,27,28,29]). Because these metabolites have been the last to be identified and measured in human blood with standardized and validated procedures [23,30,31], their interindividual variability in response to α-TOH supplementation remains unexplored. 

In this study, a targeted metabolomics procedure, recently developed for the simultaneous analysis of all the metabolites of α-TOH identified so far in human serum or plasma—that are collectively referred as to “the metabolome of vitamin E” [31]—was applied to assess the variability of vitamin E metabolism in healthy volunteers during their regular diet or after one-week supplementation with a supra-nutritional dosage (800 U/day) of the natural form of the vitamin (RRR-α-TOH). Metabolomics data will include the levels of the free form of polyunsaturated fatty acids (PUFAs) and their metabolites that have been included in the one-shot LC-MS/MS analysis protocol recently developed in our labs to assess the metabolome of vitamin E (VE) [30]. Leukocyte pregnane X receptor (PXR) and CYP4F2 expression will be investigated by immunoblot, these two being molecular targets of α-TOH with a proposed role in the metabolism and function of this vitamin [26].

## 2. Materials and Methods

### 2.1. Study Design

This intervention trial was carried out within the grant program of the Italian Ministry of University and Research: National Technology Agrofood Cluster, Health and Nutrition area—PROS.IT project (CTN01_00230_413096). The study was approved by the Bioethics Committee of the University of Perugia.

According to the findings reported in previous metabolomics studies by us [30,32] and others [21], a number of 15 participants was calculated to be sufficient to target the hypothesis that a significant variation of metabolite levels can be achieved upon α-TOH supplementation in healthy individuals. As a consequence, twenty healthy volunteers eligible for the nutritional intervention were identified among the personnel of the University of Perugia and the research staff of this study, and, after verification of exclusion criteria (absence of diagnosis of chronic-degenerative and infectious diseases), 17 subjects were included in the trial. Treatment consisted of a one-week oral administration protocol of 400 International Units (IU) of RRR-α-TOH (Abiogen Pharma Sursum, Pisa, Italy) twice-a-day in correspondence of the morning and evening meal. Blood samples were collected in ethylenediaminetetraacetic acid (EDTA)-containing vacutainer tubes before and at the end of the one-week supplementation protocol by venipuncture of the median cubital vessel that was carried out in the morning (8–9 A.M.) under fasting conditions. Blood samples were immediately processed for separation of plasma and mononuclear leukocytes that were stored at −80 °C for no longer than two weeks before utilization in the experimental procedures. 

Characteristic of participants, including anthropometric parameters [age, weight, body mass index (BMI), waist circumference (WC)] are shown in Appendix A.

### 2.2. Reagents and Standards

High Performance Liquid Chromatography (HPLC)-grade solvents, beta glucuronidase from Escherichia Coli (G7396), sulphatase from Helix Pomatia (S9626), and the analytical standards α-tocopherol (α-T), d3-γ-tocopherol (d3-γ-T) and d6-αtocopherol (d6-α-T) were from Sigma Aldrich (St. Louis, MO, USA). Other analytical standards included: γ-tocopherol (γ-T) from Cognis Corporation (BASF, Ludwigshafen, Germany) and 2,5,7,8-tetramethyl-2-(2′-carboxyethyl)- 6-hydroxychroman (α-carboxyethyl hydroxychromanol, α-CEHC), d3-α-CEHC and 2, 7, 8-trimethyl-2- (2′-carboxyethyl)-6-hydroxychroman (γ-carboxyethyl hydroxychromanol, γ-CEHC) kindly gifted by Eisai Corporation (Tokyo, Japan) or purchased from Sigma Aldrich (St. Louis, MO, USA). The standards of the long-chain metabolites α-13′-(6-hydroxy-2,5,7,8-tetramethylchroman-2-yl)-2,6,10-trimethyltridecanol (13′-hydroxychromanol or α-13′-OH), α-13′-(6-hydroxy-2,5,7,8-tetramethylchroman-2-yl)-2,6,10-trimethyltridecanoic acid (13′-carboxychromanol or α-13′-COOH) were synthetized as described in Reference [33] and references therein. Deuterated and native standards of PUFAs and oxylipins were from Cayman Chemicals (Ann Arbor, MI, USA) and included: arachidonic acid (AA), arachidonic acid-d8 (d8-AA), α-linolenic acid (ALA), α-linolenic acid-d5 (d5-ALA), eicosapentaenoic acid (EPA), docosahexaenoic acid (DHA), leukotriene B4 (LTB4), leukotriene B4-d4 (d4-LTB4), 20-carboxy leukotriene B4 (20-COOH-LTB4), 20-hydroxy leukotriene B4 (20-OH-LTB4), 20-hydroxyeicosatetraenoic acid (20-HETE), 20-hydroxyeicosatetraenoic acid-d6 (d6-20-HETE), and 20-carboxy-arachidonic acid (20-COOH-AA). Individual stock and working solutions of the analytical standards were prepared in methanol and stored at −80 °C.

### 2.3. Sample Preparation for Analysis of Tocopherol, PUFAs, and Their Metabolites

Protocol for metabolites—300 µL of plasma were placed in a 15 mL polypropylene tube and spiked with 50 μL of ascorbic acid (10 mg/mL), followed by 100 μL of a solution containing beta-glucuronidase (10,000 U/mL) and sulfatase (200 U/mL) in 0.25 M sodium acetate. The sample was incubated at 34 °C for 30 min and then spiked with the internal standards d4-α-13′-COOH, d3-α-CEHC, d4-LTB4, and d6-20-HETE. After vortexing, 100 μL of glacial acetic acid and 1 mL of ethanol were added. The sample was extracted with 5 mL of a hexane/tertiary butyl methyl ether (TBME) 2/1 *v/v* containing 50 mg L^−1^ butylated hydroxytoluene (BHT). After vortexing and centrifugation, the organic layer was collected into a clean glass tube. Extraction was repeated with 5 mL of a hexane/TBME mixture (1/2 *v*/*v*) containing 50 mg L^−1^ BHT. The recovered organic phases were dried under nitrogen stream. The analytes were then dissolved in 150 μL of a mixture methanol/water (75/25 *v*/*v*) containing 1 mg L^−1^ of BHT and injected for LC-MS/MS analysis. The quantification was achieved using matrix-matched calibration standards. 

Protocol for tocopherols and PUFAs—50 μL of plasma were placed in a 15 mL polypropylene tube and spiked with the isotopologues of α-T, γ-T, AA, and ALA (d6-α-T, d3-γ-T, d8-AA and d5-ALA) that were used as internal standards. The extraction was carried out with 5 mL of a hexane/TBME mixture (4/1 *v/v*) containing 50 mg L^−1^ of BHT. After vortexing and centrifugation (3220× *g* for 5 min at 10 °C), the organic layer was collected into a clean glass tube. Extraction was repeated twice, and the recovered organic phases were dried under a stream of nitrogen and resuspended in 500 μL of methanol containing 1 mg L^−1^ of BHT. An aliquot of the extract was further diluted to assess the analytes present at highest concentrations (double injection). Compound quantification was performed by calibration curves in methanol using the internal standardization and isotopic dilution method.

### 2.4. LC-MS/MS Analysis

The validated targeted metabolomics protocol described in Reference [30] was implemented in this intervention study to assess at the same time the main vitamers and the different classes of metabolites that characterize the metabolome of vitamin E. The method has also been adapted to the simultaneous analysis of the main PUFA species and some of their eicosanoid products. Briefly, liquid chromatography separation and mass spectrometry detection were performed on a Finnigan Surveyor LC pump system combined with a triple quadrupole mass spectrometer (TSQ Quantum Ultra, Thermo Fisher, Palo Alto, CA, USA). The separation of metabolites was achieved using a Gemini C18 column (100 mm × 2.0 mm, 3.0 μm, 100 Å, Phenomenex, Torrance, CA, USA) and water (A) and methanol (B) as mobile phases, both containing formic acid (0.1%). For the separation of tocopherols/PUFAs, eluent A was water with 0.01% of formic acid and eluent B methanol, both containing ammonium formate (0.1 mM). The separation gradient was initiated with 50% eluent B for 1 min. followed by a linear increase up to 100% B in 8 min; this condition was maintained for 7 min. Finally, the system returned to 50% B in 1 min and was re-equilibrated for 8 min. The column temperature was 40 °C and the sample temperature was 12 °C. The flow rate was 0.3 mL min^−1^ and the injection volume 5 μL. The electrospray ionization source (ESI) operated in positive mode for the analysis of vitamin E compounds and in negative mode for the PUFA-related molecules.

### 2.5. Immunoblot

Peripheral blood mononuclear cells (PBMLs) were isolated using Lympholyte-H (Cedarlane Laboratories, Ontario, Canada). To extract PBML proteins, the cells were incubated for 40 min at 4 °C in lysis buffer (Cell Signaling Technologies, Denver, MA, USA) supplemented with protease and phosphatase inhibitor mixture (Pierce, Thermo Fisher Scientific, Waltham, MA, USA) and fresh 1 mM phenylmethylsulfonyl fluoride (PMF, Sigma-Aldrich, MO, USA). After incubation, the samples were centrifuged (14,000 rpm for 30 min at 4 °C), and the supernatants were collected for immunoblot analysis. Total proteins of cell lysates were quantified by bicinchoninic acid (BCA) assay (Thermo Fisher Scientific). Immunoblot of PXR was performed by protein separation on 12% SDSPAGE and subsequent electroblotting to a nitrocellulose membrane (Thermo Fisher Scientific). After blocking with 5% nonfat milk, the membrane was incubated with anti-PXR antibody (bs-2334R; 1:500, Bioss antibodies), anti-CYP4F2 (1:500, Santa Cruz Biot., Santa Cruz, CA, USA), and anti β-actin (#4967, 1:1000, Cell Signaling Technologies, CST, Denvers, MA, USA) and then with a horseradish peroxidase-conjugated secondary antibody (1:2000, Cell Signaling Technologies). Band detection was carried out by enhanced chemiluminescence (ECL)-plus (Pierce, Thermo Fisher Scientific) according to the manufacturer’s instructions. Images of were analyzed with “Image J” software.

### 2.6. Statistical Analysis

Data are presented as mean ± standard deviation (SD). Considering the differences in data distribution before and after supplementation, both parametric and non-parametric methods were applied to assess differences within and between groups of data obtained in these two time-points of the intervention study. One-way Analysis of variance (ANOVA) followed by Bonferroni post-hoc test was utilized as reference statistics protocol, and linear regression analysis and multivariable correlations were assessed utilizing the statistics package of GraphPad Prism version 5.0 (San Diego, CA, USA). *p* < 0.05 was considered significant.

## 3. Results

### 3.1. Levels and Interindividual Variability of α-Tocopherol, γ-Tocopherol and Their Metabolites

After supplementation, mean concentrations and coefficient of variation (CV) of α-TOH increased by approximately 2 and 4 times the baseline values, respectively (Table 1 and Figure 2). Such variability of α-TOH was completely corrected after normalization of the vitamin levels for the concentrations of cholesterol (ratio α-TOH/cholesterol). The latter is a major component of circulating lipoproteins, i.e., the tissue transferring system of vitamin E; therefore, such normalization helps to identify possible confounding factors deriving from both physiological and pathological variations of the lipid status (for example, unreported food intake, subclinical forms of dyslipidemia with mild hypercholesterolemia or non-alcoholic fatty liver) (extensively review elsewhere in References [5,26]).

Although no significant difference was observed between the two sexes, female showed a trend toward lower levels of α-TOH at baseline and a higher increase of these levels after supplementation compared with man (Figure 2).

The supplementation of α-TOH significantly reduced the levels of plasma γ-TOH, which is a well-known effect of the administration of supra-nutritional dosages of this vitamin in humans [34]. The subjects’ sex did not influence this response nor the baseline levels of γ-TOH. 

All the enzymatic metabolites with different side-chain length, and α-TQ that was investigated as indicator of α-TOH auto-oxidation, showed higher concentrations after supplementation; in all these metabolites, no significant differences between male and female subjects were observed. 

Considering the coefficient of variation (CV%) and the extent of variation after supplementation, some specificities in the different components of this metabolome can be identified. The response to supplementation followed the order of magnitude (expressed as fold-increase of the mean plasma concentrations after supplementation over the baseline levels): α-13′-COOH < M3 < α-TQ < α-13′-OH < M2 < α-CMBHC < M1 < α-CEHC. γ-CEHC concentrations showed a trend toward an increase after supplementation that was associated with a decrease of its precursor γ-TOC. 

Among the compounds with higher response, α-CEHC, M1, and especially α-13′-OH, were characterized by the lowest CV values after supplementation. On the contrary, the response of α-CMBHC was associated with a marked increase of the CV value.

### 3.2. Confounding Variables and Correlations

Anthropometric parameters (age, BMI, and WC) were assessed as possible confounding factors (Appendix A). The effect of these variables on data distribution was evaluated utilizing a linear correlation model in which baseline and post-supplementation data were compared. A significant positive correlation was observed when baseline α-TOH was matched with age (Appendix A; R^2^ = 0.28, *p* < 0.05). This correlation was not significant after supplementation by the increased variability of α-TOH levels. The same trend toward a positive correlation of baseline α-TOH levels was observed in the case of WC (R^2^ = 0.17, *p* = 0.09). 

No significant correlations with age were observed for all the metabolites investigated either before or after α-TOH supplementation (Appendix A). Significant correlations of BMI and WC were reported for some metabolites (Appendix A), including a negative correlation of BMI with baseline levels of α-CMBHC (R^2^ = 0.41, *p* < 0.01) and post-supplementation levels of M2 (R^2^ = 0.25, *p* < 0.05), whereas WC was positively correlated with baseline levels of α-CMBHC (R^2^ = 0.58, *p* < 0.01) and post-supplementation levels of M2 (R^2^ = 0.26, *p* < 0.05) and M3 (R^2^ = 0.29, *p* < 0.05).

Multiparameter regression analysis data of the different metabolites and α-TOH levels determined before and after supplementation are shown in Appendix A. α-TQ was the only metabolite to show a significant positive correlation after supplementation with the levels of its precursor α-TOH (Figure 3, left panel; R^2^ = 0.25, *p* < 0.05), whereas M1 correlated with the cholesterol-corrected levels of α-TOH at baseline evaluation (Figure 3, right panel; R^2^ = 0.48, *p* < 0.01).

### 3.3. PUFA Analysis 

The levels of PUFAs in the free fatty acid fraction of plasma represent an indicator of optimal nutrition and are the target of the antioxidant activity of vitamin E [35]. As a consequence, their levels were measured in this study utilizing a recently-developed metabolomic method that simultaneously determines in the same run these fatty acids and all the metabolites of vitamin E [30,36]. All plasma levels of PUFAs showed a trend toward increased levels after α-TOH supplementation (Appendix A). For these fatty acids, interindividual variability of data was remarkably high both before and after supplementation and no any significant correlation was observed for these species with the levels of α-TOH and its metabolites in plasma. 

### 3.4. Molecular Studies

The expression of PXR, but not that of CYP4F2, increased after α-TOH supplementation (Figure 4 and Appendix A). PXR expression showed the same levels of variability before and after supplementation (Appendix A), and linear regression analysis data demonstrate a significant positive correlation between the basal levels of the α-TOH/Cholesterol ratio and PXR levels measured either before or after supplementation (Appendix A).

Moreover, post-supplementation levels of M1 positively correlated with PXR (R^2^ = 0.295, *p* < 0.05; Appendix A), whereas all the other metabolites did not correlate with the levels of this nuclear receptors either before or after supplementation (not shown).

## 4. Discussion

The metabolism and function of vitamin E are characterized by a marked interindividual variability, affecting for instance blood levels, antioxidant effects and biotransformation rate. Such variability was investigated for the first time in this vitamin E supplementation study as far as the entire series of α-TOH metabolites identified to date in human blood is regarded, the so-called “vitamin E metabolome”. The possibility to study this metabolome in human plasma has only recently been achieved by the development of targeted metabolomics methods that have specifically been validate for this application [30,32,36].

We also investigated for the first time in this study the effect of α-TOH supplementation on two possible molecular targets of vitamin E in human tissues, namely PXR nuclear receptor, which is considered to represent a master regulator of VE metabolism [33,37], and CYP4F2, a putative tocopherol ω-hydroxylase [38]. 

Formerly, all metabolites increased their concentrations in response to α-TOH supplementation; however, the response of the different metabolites was heterogeneous (see CV% and fold-increase data of Table 1) and independent from the concentrations of their precursor α-TOH. The only exception to this general observation was the free radical-derived metabolite α-TQ that showed a significant correlation with α-TOH concentrations at the end of the supplementation protocol. These findings indicate that the different components in the enzymatic branch of vitamin E metabolism are very susceptible to biological heterogeneities, possibly by the participation of different groups of genes and proteins (recently reviewed in Reference [26]). On the contrary, the free radical mediated metabolism of this vitamin to form α-TQ appears to be a less variable process of human tissues, which is consistent with previous in vitro data of α-TOH supplementation obtained in human liver cells [36].

Second, the different degrees of variability observed for the response of some metabolites, as measured by the CV (SD*100/mean value), highlight the intervention of individual, and so far unknown, factors that affect the different steps of formation and clearance of these metabolites. These steps depend on the expression and activity of CYP450 isoenzymes, dehydrogenases, β-oxidation enzymes, and transporters [39], and their investigation by metabolite analysis may help to shed light on the genetic variability alleged to explain individual variations in the absorption and biotransformation of vitamin E to CEHC metabolites [21]. In our study, α-CEHC is the enzymatic metabolite the mean levels of which showed the highest response to supplementation (>20 times the baseline levels) and a CV% that remained unchanged during the supplementation study. Similar data were observed for M1, one of the unknown LCMs first identified in these laboratories as the most abundant biotransformation product of α-TOH in human tissues [30,32], while the highly responsive middle-chain metabolite α-CMBHC was associated with a marked increase of CV after supplementation. Among the LCMs, also α-13′OH and M2 were characterized by marked increases of their concentrations, but only in the case of α-13′OH such response to α-TOH supplementation was associated with a decreased variability. These results confirm the possibility to use α-CEHC as a biomarker of vitamin E intake, already proposed in pioneering studies in which this metabolite was investigated in urine [40] and plasma [41]. Moreover, based on present data, we suggest that this role could be extended to M1 and also to α-13′OH, which is also important to monitor the α-TOH bioactivation process [27,39]. In fact, α-13′OH is one of the bioactive derivatives of the enzymatic processing of α-TOH with proposes anti-inflammatory function [42] and recent studies also suggested agonist activity of this LCM on human liver peroxisome proliferator-activated receptor gamma (PPARγ) [43] and on the PPARγ-apolipoprotein E (APOE) axis of mouse astrocytes exposed to β-amyloid peptide toxicities [44]. α-13′OH is also the direct precursor of α-13′COOH that is reported to be a potent lipoxygenase-5 (LOX-5) inhibitor [45]. However, α-13′COOH showed the lowest levels of upregulation after supplementation among the entire series of metabolites investigated in this study (approximately 5 times lower compared with α-13′OH), suggesting a rapid transformation of this acid derivatives of α-TOH throughout the enzymatic pathway.

Importantly enough, the interindividual variability of α-TOH levels after supplementation, was reduced upon correction for cholesterol levels (indicated as α-TOH/Cholesterol ratio in Figure 2 and Table 1), confirming the close relationship of vitamin E metabolism with lipoprotein metabolism [5,26]. Alternatively, the baseline variability of α-TOH levels was affected to some extent by the subject age and anthropometric characteristics, mainly WC. However, all these factors and the same variability of α-TOH levels did not appear to influence the formation of the bioactive metabolite α-13′OH, as well as of all the other enzymatic metabolites. 

The fact that blood lipids, subject age, and WC are among the factors that may contribute to the interindividual variability of α-TOH levels and metabolism is not surprising. In fact, the age-related decline of physiological functions and the excess of fat depots appear to play a key role in determining the status and systemic availability of this vitamin [46], and highly prevalent ailments, such as obesity, metabolic syndrome, and non-alcoholic fatty liver disease, are all associated with sequestration, and impaired catabolism and turnover of tissue α-TOH [47,48,49]. Subclinical forms of these metabolic conditions (such as overweight, benign obesity, and fatty liver) are very common in apparently healthy people and could interfere with vitamin E metabolism, stimulating its free radical-mediated oxidation [36] and/or leading to reduced biotransformation throughout enzymatic pathways [48], thus representing potential factors of variability of this metabolome that are worth investigating in future clinical trials. Considering the anthropometric characteristics and baseline levels of α-TQ and lipid corrected levels of α-TOH, the presence of subclinical conditions of fatty liver [36] can be excluded in the heathy volunteers of this study. 

α-TOH supplementation was confirmed to interfere with the plasma levels of γ-TOH [50,51,52,53,54], which could be explained, at least in part, by the increased biotransformation of this vitamer to γ-CEHC. However, also in this case, the interindividual variability of metabolomics data markedly interfered with the possibility to observe significant correlations between the upregulation of α-TOH levels and the changes of γ-TOH and γ-CEHC levels. 

Exploring individual factors that may affect the variability of metabolomics data at the molecular level, PXR protein, but not CYP4F2, expression significantly increased by the effect of the supplementation protocol, and baseline PXR showed significant correlations with α-TOH/Cholesterol levels measured either before or at the end of the supplementation protocol; moreover, PXR data maintained the same interindividual variability throughout the supplementation study. Worthy of note is that this is the first time that this nuclear receptor is investigated in humans as an indicator of the metabolic response to α-TOH supplementation. Although the small number of subjects investigated is a major limit in this study to attain conclusive data, these correlations confirm the proposed role of PXR as a molecular target of vitamin E [33,37]. These findings also suggest great potential for the combined determination of PXR expression in PBML and metabolite levels in plasma, as a strategy to predict at the individual level the nutritional and biotransformation response to α-TOH in a wide range of intakes. The poor relevance of CYP4F2 in the human metabolism of vitamin E proposed in other reports [49,55] is once more supported by the experimental data of this study.

## 5. Conclusions

In conclusion, the present study describes for the first time the interindividual variability that the different metabolites of α-TOH present during the supplementation of this vitamin in healthy humans. Such original information has been obtained utilizing validated protocols that allow metabolite quantitation over a wide range of concentrations [23,30,32]. The investigated metabolites include molecules that have been reported to have important biological roles. More in detail, the LCMs α-13′OH and α-13′COOH have been described to represent ligands and potent modulators of nuclear receptors and transcription factors (such as PXR and PPARγ), as well as of enzymatic proteins involved in physiological processes, such as eicosanoid metabolism, regulation of inflammatory pathways, lipid metabolism and detoxification [26,27,29]. Metabolites assessed in this intervention study also include α-TQ which is a promising in vivo indicator of lipid peroxidation [36], and some isomeric forms of α-13′OH and α-13′COOH (namely M1, M2, and M3), recently identified in human plasma as products of the in vivo biotransformation of α-TOH [30,32]. Worthy of note is that M1 is the most abundant LCM detected in this metabolome and it was the only metabolite that positively correlated with baseline levels of α-TOH. The molecular identity of these recently identified LCMs is now under investigation.

Further studies are in progress in our laboratories to shed more light on the causal relationship between the gene expression and the metabolic (enzymatic) response that PXR recruits at the individual level upon α-TOH supplementation. These include the analysis of CYP3A4, the expression of which cannot be evaluated with the samples available in the present study. Future nutrigenomics approaches should also take into due consideration the polymorphic characteristics of the different groups of genes involved in the uptake and metabolism of this vitamin. Understanding these aspects will also help to better investigate α-TOH biotransformation and its supposed role as physiological bioactivation process of human tissues [27,39].

## Figures and Tables

**Figure 1 antioxidants-10-00173-f001:**
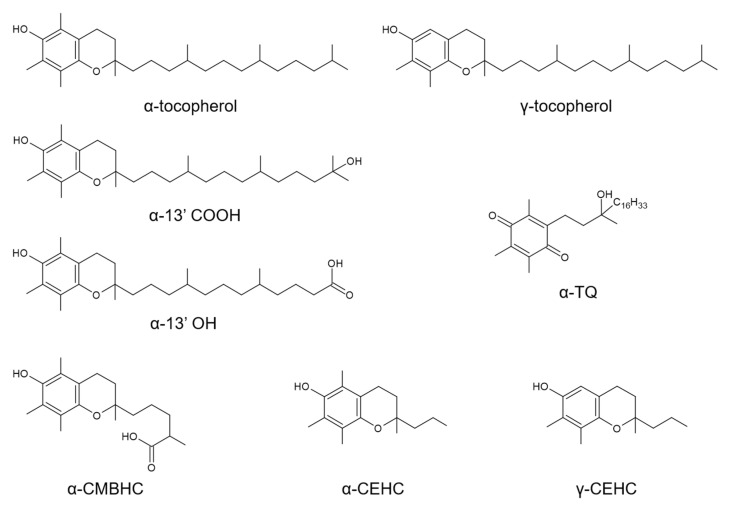
Chemical structure of the vitamin E compounds included in this metabolomics investigation. These include the vitamers α-tocopherol and γ-tocopherol, the enzymatic long-chain metabolites (LCMs) α-13’hydroxychromanol (α-13′OH) and 13′-carboxychromanol (α-13′COOH), the middle-chain metabolite (MCM) 2,7,8-trimethyl-2-(δ-carboxymethylbutyl)-6-hydroxychroman (α-CMBHC), the short-chain (SCMs) carboxy-ethyl-hydroxychroman metabolites (α-CEHC and γ-CEHC, and the free radical-derived LCM α-tocopheryl quinone (α-TQ).

**Figure 2 antioxidants-10-00173-f002:**
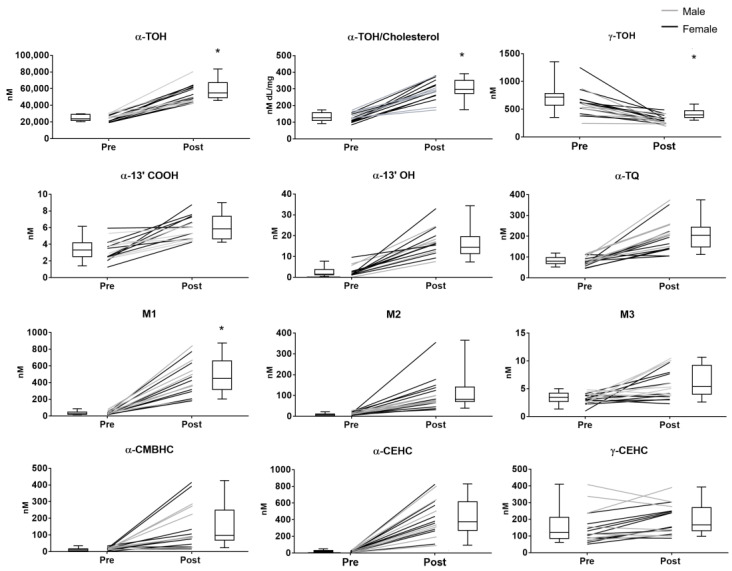
Plasma levels of the vitamin E compounds measured in healthy subjects before (pre) and after (post) supplementation with RRR-α-TOH (α-tocopherol). * *p* < 0.05. Black line = female, grey line = male.

**Figure 3 antioxidants-10-00173-f003:**
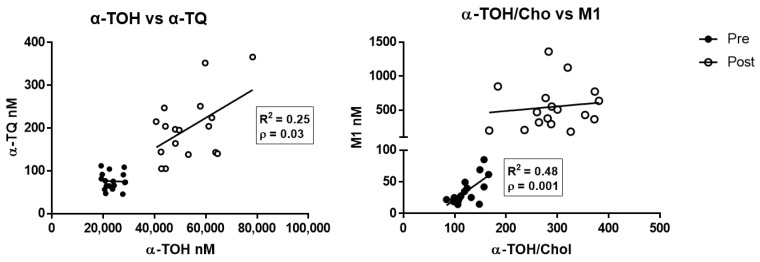
Correlation between α-TOH and the free radical-derived metabolites α-TQ (**left**), and between cholesterol-corrected level of α-TOH and M1 metabolite (**right**), measured in healthy subjects before (pre) and after (post) supplementation with RRR-α-TOH.

**Figure 4 antioxidants-10-00173-f004:**
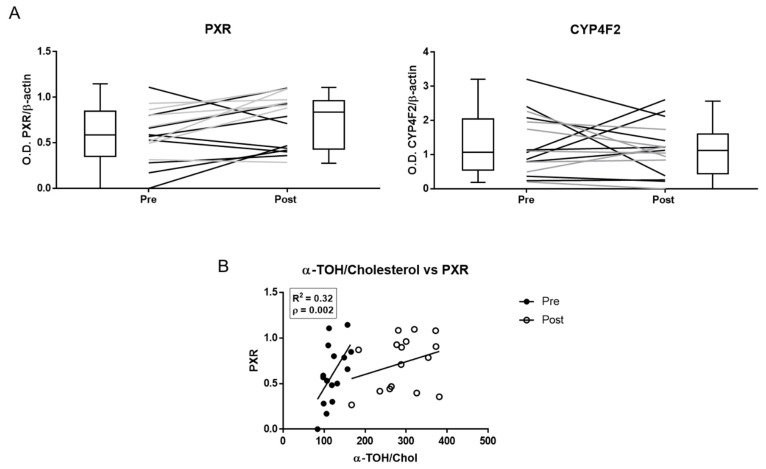
Pregnane X receptor (PXR) and CYP4F2 protein expression in peripheral blood mononuclear cells (PBMLs) of healthy subjects measured by immunoblot before (pre) and after (post) α-TOH supplementation. (**A**) Densitometric data of band analysis are expressed as optical density units. Blotting images are shown in Appendix A. (**B**) Correlation between the plasma levels of α-TOH and PXR expression in PBMLs.

**Table 1 antioxidants-10-00173-t001:** Statistical parameters and data distribution analysis.

		Mean	SD	Median	Range	Variance	Coefficient of Variation (%)	Fold Increase
α-TOH (nM)	Pre	23,483.4	3450.9	22,434.6	19,181–28,618	11,208,207.3	14%	2.27
	Post	53,237.9	10,530.0	49,653.6	40,656–78,248	104,358,536.9	43%	
α -TOH/Cho ^#^	Pre	123	24.8	119	84–166	577.5	20%	2.36
	Post	291	60.9	288	167–381	3486.61	20%	
γ-TOH (nM)	Pre	615.2	228.1	614.5	254–1250	48,968.6	38%	0.51
	Post	315.6	79.2	297.4	200–489	5909.1	13%	
α-TQ (nM)	Pre	75.8	20.1	73.5	45–111	380.0	26%	2.63
	Post	199.7	74.9	196.6	104–366	5275.1	97%	
α-13’COOH (nM)	Pre	3.5	1.8	3.1	1–8	3.0	49%	1.66
	Post	5.8	1.4	5.6	4–8	2.0	39%	
M1 (nM) ^§^	Pre	35.1	20.4	25.6	14.3–85	390.8	56%	15.65
	Post	549.2	329.6	471.4	182–1363.3	102,250.2	58%	
M2 (nM) ^§^	Pre	11.5	6.3	11.2	4.5–24.7	37.2	53%	8.89
	Post	102.3	76.8	73.9	31.5–355.9	5556.7	73%	
α-13’OH (nM)	Pre	2.6	2.1	1.7	0.4–6.9	4.1	86%	7.23
	Post	18.8	8.7	16.8	8–41.5	70.6	46%	
M3 (nM) *	Pre	3.0	0.9	3.0	1–4.6	0.8	31%	1.97
	Post	5.9	2.8	5.1	2.3–10.3	7.2	46%	
α-CMBHC (nM)	Pre	11.0	7.3	11.6	0.6–27.6	49.9	66%	11.29
	Post	124.2	107.4	84.8	15.7–392.8	10,818.2	86%	
α-CEHC (nM)	Pre	20.3	11.6	19.0	6.2–44.6	126.7	57%	20.83
	Post	422.8	224.2	371.0	93–824.7	47,121.9	53%	
γ-CEHC (nM)	Pre	102.1	37.5	102.2	50.9–172.9	1297.7	37%	1.69
	Post	172.7	77.1	153.1	87.4–303.5	5571.9	45%	
PXR	Pre	0.6	0.3	0.57	0–1.14	0.09	53%	1.20
	Post	0.72	0.29	0.82	0.2–1.0	0.07	39%	
CYP4F2	Pre	1.30	0.88	1.09	0.2–3.2	0.73	68%	0.9
	Post	1.17	0.75	1.18	0–2.2	0.52	64%	

Cho: Cholesterol. Coefficient of variation (%) is calculated using the formula: (SD/average) * 100. ^#^ In the α-TOH/Cholesterol ratio, α-TOH levels were in nM and cholesterol levels were in mg/dl. ^§^ Quantification was in nmol/l of α-13’COOH equivalents; * Quantification was in nmol/l of α-13’OH equivalents.

## Data Availability

The data presented in this study are available on request from the corresponding author.

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
