# Peer review of "Alpha-Tocopherol Metabolites (The Vitamin E Metabolome) and Their Interindividual Variability during Supplementation"

_antioxidants, 2021, doi:10.3390/antiox10020173_

Round 1
Reviewer 1 Report
Congratulations on an interesting idea for a patent and article.
The subject matter presented in the article by the team of authors is one of the most important topics that appear in publications today. I confirm that the presentation of the results of this type of research is the first time in publications I know. The paper presents a complete picture of changes in the α -TOH metabolome before and after supplementation of this vitamin in healthy people.
In my opinion, the results of future research on describing the causal relationship between gene expression and the metabolic (enzymatic) response that PXR is recruited on an individual level after α-TOH supplementation may also be interesting. This includes the analysis of CYP3A4, expression of which cannot be assessed from the samples available in this study. Future nutrigenomic approaches should also take due account of the polymorphic features of the different groups of genes involved in the uptake and metabolism of this vitamin.
The team of authors reviewed specialist literature related to the topic of the article. The conclusions drawn are interesting for researchers of this issue.
There are small errors in this article that should be corrected:
- Figures 2 and 4 are not legible, due to the rather important information they contain, their legibility should be improved.
- Too long sentences appear in the article, they should be corrected.
Author Response
Congratulations on an interesting idea for a patent and article.
The subject matter presented in the article by the team of authors is one of the most important topics that appear in publications today. I confirm that the presentation of the results of this type of research is the first time in publications I know. The paper presents a complete picture of changes in the α -TOH metabolome before and after supplementation of this vitamin in healthy people.
In my opinion, the results of future research on describing the causal relationship between gene expression and the metabolic (enzymatic) response that PXR is recruited on an individual level after α-TOH supplementation may also be interesting. This includes the analysis of CYP3A4, expression of which cannot be assessed from the samples available in this study. Future nutrigenomic approaches should also take due account of the polymorphic features of the different groups of genes involved in the uptake and metabolism of this vitamin.
The team of authors reviewed specialist literature related to the topic of the article. The conclusions drawn are interesting for researchers of this issue.
There are small errors in this article that should be corrected:
- Figures 2 and 4 are not legible, due to the rather important information they contain, their legibility should be improved.
- Too long sentences appear in the article, they should be corrected.
Answer: we are indebted with this Reviewer for the positive comments and consideration for our work. Both the two points raised by the Reviewer have been considered during the revision of our manuscript and extensive rephrasing of some pats and reorganization of the entire manuscript should have improved the language and style of this article.
Reviewer 2 Report
English language of the paper requires substantial overall improvement. Introduction and discussion sections need to be substantially revised, to make the goal, experimental design, outcomes and significance of the research much clearer.
Revise main title and the running title.
Abstract:
Metabolome of vitamin E?? This definition needs to be revised or better explained. Accepted definition of metabolome - 'the total number of metabolites present within an organism, cell, or tissue', so it is not clear what 'metabolome of vitamin E' is.
''Explore the metabolome of αtocopherol (a-TOH, vitamin E) in a comprehensive and thorough manner, whereas nutrigenomic studies identified the biological properties of the different components in this metabolome" - again, revise the wording, the meaning of the phrase is nor clear.
Keywords: write out what PPARg means.
Introduction:
Explain what ‘human LDL protection’ is.
'Long-chain metabolites (LCMs; Figure 1)' - LCMs are not marked in Fig 1.
Explain what RRR-a-TOH is. Any abbraviation introduced into the text for the first time needs to be written out.
What is 'entire spectrum of VE metabolites in human serum or plasma'? Further clear explanation is needed.
Overall, introduction section is quite confusing. It was hard to figure out what exactly will be analyzed as part of VE metabolome and what is the significance of this investigation.
M&M section
'Healthy volunteers were enrolled among the research staff of the institutions participating to this study'. not sure what this means.
National Agrofood Technology Cluster program – reference or further description of the program is needed
400 UI – what is UI?
Abiogen Pharma Sursum – City? Country?
Reagents and standards - explain why those were used in this study. Separate them into distinct groups.
Explain 'PUFA' abbreviation in the text.
Sample preparation Tocopherol and PUFA metabolites – modify the heading
kiquid chromatography – correct the typo
Statistical analysis:
‘Data were as mean ± S.D’ – reword the sentence.
Results
‘this increase was associated with a 4-fold higher CV value that was completely rectified by the correction α-TOH for cholesterol concentrations’ (Table 1 and Figure 2) – not sure what this means
‘cholesterol-corrected level of a-TOH and M1 metabolite’, - explain how cholesterol correction works
For how long did the metabolite increase after supplementation last?
Discussion section needs to be revised as well for clarity.
Author Response
English language of the paper requires substantial overall improvement. Introduction and discussion sections need to be substantially revised, to make the goal, experimental design, outcomes and significance of the research much clearer.
Answer: the manuscript has thoroughly been revised by an expert in scientific English. Introduction and discussion sections have been revised according with the suggested need of a more punctual description of the project goal and experimental design. Changes are highlighted in the text for reviewer consultation and all we hope that the proposed revision may satisfy the requests of this Reviewer.
Revise main title and the running title.
Answer: main title and the running title have been revised as suggested
Abstract:
Metabolome of vitamin E?? This definition needs to be revised or better explained. Accepted definition of metabolome - 'the total number of metabolites present within an organism, cell, or tissue', so it is not clear what 'metabolome of vitamin E' is.
''Explore the metabolome of αtocopherol (a-TOH, vitamin E) in a comprehensive and thorough manner, whereas nutrigenomic studies identified the biological properties of the different components in this metabolome" - again, revise the wording, the meaning of the phrase is nor clear.
Answer: We appreciate the request of clarification by the Reviewer.
Definitions such as “Metabolome of Vitamin E” or “''Explore the metabolome of α-tocopherol …” could be justified by the quite common utilization of this terminology in the scientific literature, see for example: “The metabolome of [2-14C](−)-epicatechin in humans…” Ottaviani J.I., et al. Sci Rep 2016; “The serum vitamin D metabolome…Tuckey RC, et al. J Steroid Biochem Mol Biol. 2019”; “The vitamin D metabolome… Jenkinson C. Cell Biochem Funct 2019”, “The Human Serum Metabolome of Vitamin B-12 Deficiency… Brito A., et al. J Nutr. 2017”, etc.
Notwithstanding, both the title and the text (abstract and other sections) have been revised according with the request of clarification explaining these definitions in more detail or changing the wording.
Keywords: write out what PPARg means.
Answer: revised
Introduction:
Explain what ‘human LDL protection’ is.
Answer: the entire paragraph has been revised and this sentence has been eliminated.
'Long-chain metabolites (LCMs; Figure 1)' - LCMs are not marked in Fig 1.
Answer: revised
Explain what RRR-a-TOH is. Any abbraviation introduced into the text for the first time needs to be written out.
Answer: the first mention to this acronym has been associated with the definition of “natural form of the vitamin” (see first sentence of the revised pag 4). RRR indicates the configuration of the three chiral centers of the side chain.
What is 'entire spectrum of VE metabolites in human serum or plasma'? Further clear explanation is needed.
Answer: the sentence has been revised.
Overall, introduction section is quite confusing. It was hard to figure out what exactly will be analyzed as part of VE metabolome and what is the significance of this investigation.
Answer: we revised this section in order to explain better what was analyzed and the scope of this study. The same has been done in the Abstract.
M&M section
'Healthy volunteers were enrolled among the research staff of the institutions participating to this study'. not sure what this means.
Answer: the sentence has been eliminated and the definitions of the study participants is now reported in the second sentence of the paragraph.
National Agrofood Technology Cluster program – reference or further description of the program is needed
400 UI – what is UI?
Abiogen Pharma Sursum – City? Country?
Answer: all these sentences have been revised according with the suggested changes or requests of clarifications.
Reagents and standards - explain why those were used in this study. Separate them into distinct groups.
Answer: the suggested changes were implemented
Explain 'PUFA' abbreviation in the text.
Answer: the acronym is now written out (see first sentence of the revised pag. 4).
Sample preparation Tocopherol and PUFA metabolites – modify the heading
Answer: the heading has been revised
kiquid chromatography – correct the typo
Answer: revised
Statistical analysis:
‘Data were as mean ± S.D’ – reword the sentence.
Answer: revised
Results
‘this increase was associated with a 4-fold higher CV value that was completely rectified by the correction α-TOH for cholesterol concentrations (Table 1 and Figure 2) – not sure what this means
‘cholesterol-corrected level of a-TOH and M1 metabolite’, - explain how cholesterol correction works
Answer: the two sentences have been revised and the lipid correction
For how long did the metabolite increase after supplementation last?
Answer: we agree with the reviewer that this is an intersecting aspect in this type of intervention studies. Unfortunately, we did not plan to address this in our study. However, the kinetics of vitamin E metabolism in humans was already investigated by some of us in the past utilizing deuterium-labeled vitamers (see for example Galli F. et al. Free Rad Biol Med 2002) and also by other authors later on (see for example Violet et al JCI Insights 2020). Vitamin E has got a rapid metabolism and typically in single bolus oral administration protocols, the levels of the final compounds in the enzymatic metabolism of the vitamin (i.e. CEHCs) get back to baseline levels in few hrs (usually within 6-12 hrs) and their levels peak within 1 hr post-supplementation for the non-alpha forms, whereas the alpha-tocopherol has got a slightly slower metabolism (peaking between 1 and 2 hrs). Of course, in the present study the long-lasting supplementation with high dosages of the vitamin may lead to some hepatic and extra-hepatic accumulation of the vitamin, thus leading to possible prolonged formation and blood transferring of the different metabolites. May be there will be occasions in the future to explore this important aspect of vitamin E metabolism. Thank you for advising.
Discussion section needs to be revised as well for clarity.
Answer: revised
Reviewer 3 Report
In the manuscript “The metabolome of alpha-tocopherol and its interindividual variability in healthy subjects” Dr Bartolini and coworkers investigated the metabolites presents before and after supplementation of vitamin E in healthy humans by using an original approach. The authors correlated the presence of several α-TOH-derived metabolites with the subject age (and sex) and their anthropometric characteristics (i.e., BMI and WC). Moreover, they suggested the possibility to use the levels of M1 and α-13’OH metabolites, and the of PXR protein (respectively in the blood or PBML cells) as new markers to monitor α-TOH bioactivation. The manuscript has a clear rationale, it is well conducted and the results are innovative and original.
The manuscript overall is well written, despite the authors should pay more attention in writing the abstract, which is not well organized and is not clear in the hypothesis, methods and results sections.
Major concerns
1) The major limit of the manuscript is the small number of the subjects enrolled in the study. The small scale of the study automatically excludes the possibility to obtain significant differences or correlations among the investigated parameters, even if they were present. The reviewer believes that the authors should stress this limitation including a short discussion about it.
2) The value of BMI and WC should be reported for each subject before and after the period of vitamin E supplementation. Are they changed after supplementation?
2) Can the authors obtain a correlation between the quantity of lean mass and the presence of different metabolites in the blood of subjects after the period of the vitamin E supplementation?
3) The authors should consider the age-related decline of skeletal muscle mass in the analyses and in the interpretation of their results. It is known that α-TOH is recovered in the skeletal muscle other than in the liver and adipose tissue. The quantity and the quality of myofibers might influence the variability of α-TOH-derived metabolites.
Minor points:
1) the authors should enlarge the stars in Figure 2.
2) the authors should better explain the results achieved after the statistical analyses by reporting along the text the values of correlation coefficients, for example.
Author Response
In the manuscript “The metabolome of alpha-tocopherol and its interindividual variability in healthy subjects” Dr Bartolini and coworkers investigated the metabolites presents before and after supplementation of vitamin E in healthy humans by using an original approach. The authors correlated the presence of several α-TOH-derived metabolites with the subject age (and sex) and their anthropometric characteristics (i.e., BMI and WC). Moreover, they suggested the possibility to use the levels of M1 and α-13’OH metabolites, and the of PXR protein (respectively in the blood or PBML cells) as new markers to monitor α-TOH bioactivation. The manuscript has a clear rationale, it is well conducted and the results are innovative and original.
The manuscript overall is well written, despite the authors should pay more attention in writing the abstract, which is not well organized and is not clear in the hypothesis, methods and results sections.
Answer: as for the changes made according with the same criticism by the other Reviewers, the manuscript has thoroughly been revised by an expert in scientific English, and all the different sections (including abstract, introduction, methods, results and discussion) have been revised in order to provide a more clear and punctual description of the project goal, experimental design and outcomes. Changes are highlighted in the text for reviewer consultation. All we hope that the proposed revision may satisfy the requests of this and the other Reviewers.
Major concerns
- The major limit of the manuscript is the small number of the subjects enrolled in the study. The small scale of the study automatically excludes the possibility to obtain significant differences or correlations among the investigated parameters, even if they were present. The reviewer believes that the authors should stress this limitation including a short discussion about it.
Answer: we address this point also presenting the rationale of power calculation (2nd par. In the Study design section) that was utilized during the design of this intervention trial. The number of participants needed to satisfy the hypothesis that a significant variation of metabolite levels can be observed after supplementation with alpha-tocopherol in healthy individuals is n=15.
We mention the small size of the study population and its possible impact on the conclusions drawn in this research paper (last sentence of the discussion).
- The value of BMI and WC should be reported for each subject before and after the period of vitamin E supplementation. Are they changed after supplementation?
- Can the authors obtain a correlation between the quantity of lean mass and the presence of different metabolites in the blood of subjects after the period of the vitamin E supplementation?
- The authors should consider the age-related decline of skeletal muscle mass in the analyses and in the interpretation of their results. It is known that α-TOH is recovered in the skeletal muscle other than in the liver and adipose tissue. The quantity and the quality of myofibers might influence the variability of α-TOH-derived metabolites.
Answer: this was a one-week supplementation study and not any significant change BMI and body composition can reasonably be observed during this very short time of the intervention. For that reason, we prefer to avoid the description of BMI, WC and lean mass data during such short-time intervention.
However, we really appreciate the reviewer’s suggestion to investigate the metabolism of VE in the muscle and the role of the different types of myofibers, which will be surely the subject of future investigation by this group.
Minor points:
- the authors should enlarge the stars in Figure 2.
Answer: figures have been revised accordingly
- the authors should better explain the results achieved after the statistical analyses by reporting along the text the values of correlation coefficients, for example.
Answer: the requested values of correlation coefficients have now been included in the text.
Round 2
Reviewer 3 Report
The authors reorganized the abstract and highlighted missed aspects along the text in the current version of the manuscript, which results now more clearer and organized to readers. They addressed all the points raised by the Reviewer. The manuscript has been significantly improved.